# Adaptive Fixed-Time Formation Tracking Control of USVs with Obstacle Avoidance and Prescribed Performance

Zifu LI
*College of Navigation*
*Jimei University*
Xiamen, China
fumeiscjm@jmu.edu.cn

Wenzhi Liu
*College of Navigation*
*Jimei University*
Xiamen, China
liuwenzhi@jmu.edu.cn

Yancai Hu
*School of Navigation and Shipping*
*Shandong Jiaotong University*
Weihai, China
huyancai1987@163.com

*Abstract*—With the rapid development of marine technology, Unmanned Surface Vehicles (USVs) are increasingly applied in military reconnaissance, maritime rescue, ocean mapping, and resource exploitation. These applications demand higher levels of autonomy, precision, and robustness from USVs. Particularly in complex and dynamic marine environments, the tracking control of USVs formations must simultaneously address obstacle avoidance and inter-USVs collision issues. In previous research, some scholars have introduced prescribed performance functions to ensure collision avoidance between USVs, thereby guaranteeing the system's transient and steady-state performance. Meanwhile, other scholars have employed the Artificial Potential Field (APF) method to address obstacle avoidance for both static and dynamic obstacles encountered during USVs navigation. However, these studies have not simultaneously addressed obstacle avoidance and inter-USVs collisions. Therefore, this paper proposes an adaptive fixed-time control strategy that integrates prescribed performance control and the APF method for USVs formation tracking control. This approach effectively resolves both obstacle avoidance and inter-USVs collision issues in complex environments, further enhancing the formation tracking accuracy and robustness of USVs in uncertain dynamic and unknown time-varying marine environments.

Firstly, USVs often experience complex disturbances due to model uncertainties and external environmental disturbances during navigation. To ensure the stability of the formation system, this paper designs an adaptive fixed-time disturbance observer to estimate unknown and time-varying complex disturbances. In the designed observer, an adaptive term is introduced to reduce chattering caused by the signum function, enhancing the system's smoothness and control accuracy. Secondly, to optimize the collaborative operation of USVs, particularly in obstacle avoidance and formation maintenance in complex environments, this paper incorporates an improved APF into the position error variables. Combined with prescribed performance function error transformation techniques, this approach strictly constrains the relative errors between USVs, ensuring effective obstacle avoidance and preventing collisions between USVs, thereby significantly enhancing the safety and efficiency of formation operations. Additionally, to address the singularity issue in traditional terminal sliding mode control, this paper designs a non-singular integral terminal sliding surface and innovatively introduces a segmented function. This design not only overcomes the singularity problem but also achieves real-time adjustment of the system's convergence speed through parameter adaptation in the controller. This mechanism ensures that the formation system can quickly and stably track the formation trajectory when facing complex obstacle avoidance tasks, demonstrating excellent dynamic response and adaptability. The designed controller features obstacle avoidance and collision-free operation between USVs, ensuring safe and efficient execution of formation tasks in unknown complex environments. Finally, the fixed-time stability of the closed-loop system is proven using a Lyapunov function.

To validate the effectiveness of the proposed method, we conducted multiple simulation experiments. Under various environmental conditions, we systematically evaluated the formation tracking performance, obstacle avoidance capability, and fixed-time convergence of the USVs. The experimental results demonstrate that the proposed adaptive fixed-time formation tracking control method achieves excellent formation performance under different obstacle layouts and environmental conditions, and converges stably to the target state within a fixed time. Furthermore, the method shows good robustness and flexibility in avoiding obstacles and preventing collisions between USVs.

The proposed adaptive fixed-time formation tracking control method for USVs integrates parameter adaptation technology, fixed-time control, prescribed performance control, and the APF obstacle avoidance algorithm. This approach effectively enhances the formation tracking capability and safety of USVs in complex environments. Future research could further explore control strategies in more dynamic and uncertain environments and address additional practical issues such as multi-USVs collision avoidance, communication resource optimization, and actuator faults. These efforts are expected to advance the development and application of USVs technology, providing more reliable and

**efficient solutions for marine exploration and intelligent navigation.**

*Keywords—Formation Control; Fixed-Time Control; Prescribed Performance; Obstacle Avoidance*