# OpenReview forum: "Adaptive Fixed-Time Formation Tracking Control of USVs with Obstacle Avoidance and Prescribed"
_IEEE.org/ICIST/2024/Conference — IEEE ICIST 2024 Conference Submission_

### Official Review · Reviewer_moxW · 2024-08-26
**The logic of this manuscript is clear. It is recommended to accept this paper for publication in IEEE ICIST 2024.**

**Rating:** 7
**Confidence:** 4

**Review:**

The logic of this manuscript is clear. Please answer the following review questions.

How does the proposed adaptive fixed-time control strategy that integrates prescribed performance control and the APF method compare to existing methods that address either obstacle avoidance or inter-USV collision avoidance individually?

The paper mentions that the proposed control strategy enhances the robustness of USV formations in uncertain dynamic and unknown time-varying marine environments. Can the authors elaborate more on how the adaptive nature of the control strategy contributes to this robustness?

---

### Official Review · Reviewer_DWVa · 2024-08-27
**Manuscript Accept**

**Rating:** 7
**Confidence:** 3

**Review:**

What is the main difficulty when considering simultaneously addressing obstacle avoidance and inter-USVs collisions?

Explain the features of the proposed segmented function.

How do the authors address the probable singularity issue?

---

### Official Review · Reviewer_A6F1 · 2024-08-28
**Manuscript Rejection**

**Rating:** 3
**Confidence:** 4

**Review:**

The author proposes an adaptive fixed-time control strategy that integrates prescribed performance control and the APF method for USVs formation tracking control. However, the advantages of the new control method lack depth exploration. The author need to emphasize and describe the advantages of the proposed control scheme compared to existing achievements.

---

### Decision · Program_Chairs · 2024-09-08

Accept (Oral)